# Water Footprint and Life Cycle Assessment: The Complementary Strengths of Analyzing Global Freshwater Appropriation and Resulting Local Impacts

**Winnie Gerbens-Leenes** [1,*] **, Markus Berger** [2] **and John Anthony Allan** [3]

1   Integrated Research on Energy, Environment and Society, Faculty of Science and Engineering, University of Groningen, 9747 AG Groningen, The Netherlands
2   Sustainable Engineering, Technische Universität Berlin, 10623 Berlin, Germany; markus.berger@tu-berlin.de
3   King's College London, SOAS University of London, London WC1H 0XG, UK; ta1@soas.ac.uk
*   Correspondence: p.w.leenes@rug.nl





## 1. Introduction

Considering that 4 billion people are living in water-stressed regions and that global water consumption is predicted to increase continuously [1], the analysis of water consumption and pollution along supply chains and resulting water scarcity issues is of great relevance. Since 2002, researchers from the Water Footprint (WF) community have been analyzing the global freshwater appropriation of products, companies and nations differentiating the consumption of green water (precipitation), blue water (ground and surface water) as well as gray water (theoretical amount of freshwater needed to dilute polluted water to accepted water quality standards) [2]. A few years later, water was an emerging field of research in Life Cycle Assessment (LCA), which aimed at assessing local impacts of water consumption and pollution in combination with those of greenhouse gas emissions, land use changes, etc. [3,4]. Since the beginning of these research efforts there has been a persistent debate on the orientation of the Water Footprint. In this editorial we want to shed light on the background of the methodological dispute and highlight points of disagreement but also similarities and common challenges. Further, we want to call for joint efforts from the WF and LCA communities to tackle the increasing global water challenges together. The analysis will widen the discussion by highlighting the questions that the two approaches could be trying to answer [5].

## 2. The Developments of Water Footprinting in the WF and LCA Communities

Studies on water use and scarcity in relation to consumption and trade were initiated by Tony Allan who first proposed the concept of embedded water, and later the more popular one of virtual water [6]. Allan worked on the Middle East from the mid-1960s and noted that the region had run out of water to meet its food needs by the 1970s. He also asked the question of why had the intuitively rational and much discussed outcome—water wars—not occurred? The answer was that this existential and potentially very destabilizing problem was solved by importing water-intense crops. One of the most important features of this solution was that the food commodities being imported were underpriced. The exporters, such as the United States for example, were not including the costs of their water inputs or the costs of the negative impacts on biodiversity and emissions associated with their production. Decades later they still don't. In 2002, Hoekstra [7] introduced the WF concept at an international expert meeting on virtual water trade. He showed that the WF approach is multi-dimensional. It indicated water consumption volumes and water pollution by type of pollution specified per location and in time [8]. The sustainability of water use can be assessed by comparing WFs with available water, taking environmental flow requirements into account [9]. Using the WF indicator, Mekonnen and Hoekstra [10]

quantified the gap between WFs and water availability and showed that a large part of the global population had faced water scarcity for some decades.

With an increasing application of LCA on biofuels, food and renewable raw materials, water resources have been acknowledged to be a relevant aspect that should be considered in LCA. LCA focuses on assessing environmental impacts (e.g., loss of biodiversity) resulting from environmental interferences along products' life cycles (e.g., land use change). As impacts of water resource consumption differ depending on regional scarcity and socio-economic conditions, authors from the LCA community argued for impact-based assessments [11]. As summarized by researchers from the "Water as a Global Resource" initiative [12] some of those impact assessment methods estimate the local consequences of water consumption based on freshwater scarcity [13–15]. Other methods assess the effects of water consumption on human health and well-being (due to malnutrition [13,16,17] or infectious diseases [16,18]), ecosystems (terrestrial [13,19,20], aquatic [21,22], coastal [23], wetlands [24], urban [25]), and freshwater resources [13,26,27]. The methodological enhancements and relevance of global freshwater use has led to the development of an international WF standard (ISO 14046) which specifies principles, requirements and guidelines related to WF analyses of products, processes and organizations [28].

## 3. The Scientific Dispute between Two Research Communities

As mentioned above, there has been a persistent debate on methodology between the WF and the LCA communities—mainly on the question whether the water footprint should be a volumetric or an impact-based indicator. WF scientists put the focus on water management and the volumetric analysis of water consumption and pollution, arguing that water is not only a local resource, but also a global one because water is virtually 'traded' worldwide via goods and products [2]. An analysis of local consequences of water use is considered an optional step. In contrast, the LCA community has argued that 1 m$^3$ of blue water consumption in a water scarce region is not the same as 1 m$^3$ of green water consumption in a water abundant region [29]. It claimed that volumetric footprints "have the potential to misinform" [30], and it has highlighted the necessity of an impact assessment step, which is also prescribed in the ISO standard on water footprinting [28]. Vice versa, the WF community has criticized the impact assessment methods developed in the LCA scene as a "meaningless construct" [31] or "contraproductive in reaching the Sustainable Development Goal (SDG) target 6.4" [32], especially if impact factors in LCA assessments were deployed. Methods which try to model cause–effect chains of water consumption leading to loss of biodiversity (e.g., [13]) or human health (e.g., [17]) have been criticized as "complete madness" [31].

This fundamental disagreement has led to a situation in which scientists from both communities have hardly talked to each other but about each other—in endless "reply to" paper series such as Hoekstra et al. (2009) [33] replying to Pfister and Hellweg (2009) [34] commenting on Gerbens-Leenes et al. (2009) [35], or Hoekstra and Mekonnen (2012b) [36] replying to Ridoutt and Huang (2012) [30] criticizing Hoekstra and Mekonnen (2012a) [37], etc. Considering increasing global water challenges, we feel that less energy should be wasted in such methodological battles and that the WF and LCA communities should enter more fruitful modes of discourse and cooperation. The conflict between the communities has proven to be very unhelpful. It does not for example help to answer a second question: Why is the international 'trade' in virtual water and its negative impacts on water resources, soil health, biodiversity and emissions still invisible to legislators and society? And why has this condition been firmly backgrounded [5].

## 4. Points of Agreement, Disagreement and Common Challenges Ahead

As a starting point for a constructive scientific exchange, we list the main points of agreement, as well as points on which we disagree on which both communities should try to find consensus. We also identify challenges faced by both communities if their analyses are to be policy relevant.

### 4.1. Points of Agreement

- First and foremost, WF and LCA share the same goal: the achievement of sustainable water consumption along the value chains of products and services.
- Both WF and LCA start with volumetric accounting and add a subsequent impact assessment step—the difference is the focus on volumes (WF) or impacts (LCA), which doesn't mean that the other part is meaningless.
- Methods developed in the "water footprint sustainability assessment" step (WF) can be used in the impact assessment phase of LCA and vice versa.
- Both communities highlight the relevance of spatial and temporal information and aim at increasing their resolution.
- Water pollution data generated by LCA can be applied for grey WF analysis. For example, the study of Gerbens-Leenes et al. [38] applied LCA data on water pollution for the assessment of the blue and grey water footprint of steel, cement and glass.

### 4.2. Points of Disagreement:

- Differences between the LCA and WF communities are evident in their definition of terminology. For example, the term "water footprint" is defined as "volume of freshwater used to produce goods and services" in the WF community [9] but as "metric(s) that quantify the potential environmental impacts related to water" in the LCA community [28]. This difference reflects the conflictive opinions on whether the water footprint should be a volumetric or impact-oriented indicator.
- In LCA, impact assessment is a central step in which the volumes of local water consumption are multiplied by a corresponding characterization factor, which denotes the local consequences of water consumption. In WF, the volumes are the central results and impacts can be analyzed in a subsequent step.
- Water use efficiency is not the focus of LCA [39,40] and it is assumed that water can be used without causing environmental harm in water abundant basins. In contrast, the WF approach considers that water efficiency is always important. This global perspective means that water resources should not be wasted. Water abundant basins are important as they enable the global food system to be impressively resilient but at the same time very dangerously unsustainable both economically and ecologically.
- Green water consumption is an essential part of the WF concept and often dominates a WF study. It is considered relevant as green water should be used as efficiently as possible and as green water resources used by agricultural systems are lost for local ecosystems. In LCA, green water consumption of agricultural systems is usually considered as a consequence of land-use change and impacts on biodiversity of this land-use change are already covered in the respective impact categories. Thus, green water is usually ignored as no additional impacts are seen to result from the evapotranspiration of rainwater. If at all, the change of evapotranspiration between the agricultural system and the natural vegetation, i.e., the net green water footprint [41], is considered.
- Water pollution in WF studies is assessed as the amount of water needed to dilute polluted water to accepted water quality standards, while LCA measures impacts resulting from pollutants in separate impact categories, such as eutrophication, eco-toxicity, etc.
- Focusing only on water (WF) or including water in a broader scope (LCA).

### 4.3. Common Challenges

There are six important scoping and conceptual challenges that face those proposing to carry out water resources studies deploying WF and LCA approaches. These are how to achieve the following:

- Include basin specific environmental flow requirements in the water scarcity assessments.
- Include all pollutants in agricultural WF studies, and not only focus on nitrogen.

- Assess water availability issues related to pollution.
- Access data for determining water use of products and services as well as (commercial) databases.
- Handle trade-offs between blue, green and gray water footprints but also between the water footprint and other environmental indictors (carbon footprint, land use, etc.) as well as other sustainability dimensions (social- and economic aspects).
- And finally: How can both communities take the next step from academic studies to decision relevance that supports sustainable water resource policies and practice?

## 5. Call for Action

In this special issue, researchers and stakeholders representing the WF and LCA communities are invited to submit methodological work, reviews as well as case studies to illustrate the complementary strengths of both approaches. We especially encourage researchers from both communities to cooperate and discuss the points of disagreement and challenges mentioned above in order to tackle the increasing global water challenge together.

**Author Contributions:** All authors developed the idea for this paper, accomplished the analysis and wrote the manuscript together. All authors have read and agreed to the published version of the manuscript.

**Funding:** This research received no external funding.

**Institutional Review Board Statement:** Not applicable.

**Informed Consent Statement:** Not applicable.

**Data Availability Statement:** Data sharing not applicable.

**Conflicts of Interest:** The authors declare no conflict of interest.

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
