# Peer review of "Water Footprint and Life Cycle Assessment: The Complementary Strengths of Analyzing Global Freshwater Appropriation and Resulting Local Impacts"

_water, doi:10.3390/w13060803_

Round 1
Reviewer 1 Report
Francesca Greco's work may be a good addition to the intro.
Author Response
We thank reviewer #1 who evaluates the editorial using all journal criteria as fine. The reviewer suggests to include a reference to Fransesca Greco in the Introduction. However, we prefer not to adopt this suggestion. Our Introduction gives an overview of the most important papers that contributed to the development of LCA and water footprinting. Including this reference would create the need to include more references, making the editorial too long. Moreover, Francesca Greco did her PhD work under the supervision of Tony Allen. And his work is referred to.
Reviewer 2 Report
This viewpoint and introduction to the special issue is well-written and well-organized. The argumentation is clear; differences between two research communities are touched upon and the need for cooperation made explicit. The text is fine as it is. My suggestion to the authors would be to change the order of the sections on agreements and disagreements and start with the agreements as that would follow more logically from their plea for cooperation and constructive discussion between the two research communities at the end of the preceding section. The disagreements would then be followed by the common challenges. If one would do so, some minor adaptations to the text would be needed. I am very much aware this is my personal preference and in this case I would not mind to be completely ignored ;-)
For the perfectionists among us: references #35 and #36 are the same: an article by Hoekstra and Mekonnen.
Author Response
We thank reviewer#2 for the useful suggestions to improve the paper. Indeed, it is more logical to first give the points of agreement, next the points of disagreement and finally give the common challenges. We changed this in the text.
Next, reviewer #2 mentions that reference 35 and 36 are the same. However, this should be 36 and 37. We merged these references.
Reviewer 3 Report
The manuscript well compares the different methodological approach and debate between water Foodprint and LCA communities.
There are some comments that would helpful for readers to understand the manuscript.
Firstly, a comparison table of differences between the two communities would be helpful to understand the major differences between them.
The manuscript suggests point of disagreement&agreement, and common challenge between WF and LCA communities. A figure to visualize these three points together would also make it easier for the readers to understand the important point of the viewpoint article.
Author Response
We thank reviewer #3 for the useful comments to improve the paper. It is suggested to include a table that shows the main differences and agreements of LCA and water footprinting and also add a figure. However, we listed the points of agreement, next the points of disagreement and finally give the common challenges with headings and bullet points. In this way we give a short and concise overview with some explanation. We do not think a table or figure could make this more clear.